# Ornidazole Transfer into Colostrum and Assessment of Exposure Risk for Breastfeeding Infant: A Population Pharmacokinetic Analysis

**DOI:** 10.3390/pharmaceutics15112524

**Published:** 2023-10-24

**Authors:** Sichan Li, Ming Cao, Yan Zhou, Chang Shu, Yang Wang

**Affiliations:** 1Department of Pharmacy, Wuhan Children’s Hospital, Tongji Medical College, Huazhong University of Science & Technology, Wuhan 430016, China; lisichan@zgwhfe.com (S.L.); shuchang@zgwhfe.com (C.S.); 2Department of Obstetrics and Gynecology, Wuhan Children’s Hospital, Tongji Medical College, Huazhong University of Science & Technology, Wuhan 430016, China; cm_2462006@163.com (M.C.); zhouyan@zgwhfe.com (Y.Z.); 3Office of Clinical Trial Institution, Wuhan Children’s Hospital, Tongji Medical College, Huazhong University of Science & Technology, Wuhan 430016, China

**Keywords:** breast milk, ornidazole, population pharmacokinetic, modeling, infant dose

## Abstract

Ornidazole is frequently used for the prevention and treatment of anaerobic infections after caesarean section. There is still a lack of data on the excretion of ornidazole in breast milk. Therefore, the aim of this study was to investigate the transfer of ornidazole into colostrum and to assess the risk of infant exposure to the drug via breast milk. Population pharmacokinetic analysis was conducted using datasets of plasma and milk concentrations obtained from 77 breastfeeding women to examine the excretion kinetics of ornidazole. Various factors that may affect the excretion of ornidazole were investigated. The final model was then used to simulate ornidazole concentration–time profiles in both plasma and milk. The drug exposure in body fluids and the potential risk for breastfeeding were assessed based on the safety threshold. Plasma ornidazole concentration data could be described well by a one-compartment model, and concentrations in breast milk were linked to this model using an estimated milk-to-plasma concentration ratio (MPRcon). Significant variables that influenced drug exposure and MPRcon were identified as total bilirubin levels (TBIL) and postnatal sampling time, respectively. Simulations showed that women with abnormal liver function (TBIL > 17 μmol/L) had higher ornidazole levels in plasma and milk than those with normal liver function (TBIL < 17 μmol/L), but the exposures through colostrum of lactating women from both groups were below the safety threshold. This work provides a simple and feasible strategy for the prediction of drug exposure in breast milk and the assessment of breastfeeding safety.

## 1. Introduction

Breast milk is a natural diet that contains various components, providing all the necessary nutrients for the first six months of life [1]. Breastfeeding has been found to have short-term advantages for infants in terms of improved nutrient absorption, gastrointestinal development, and immune system support. It also has long-term benefits, such as reducing the risk of obesity and diabetes in adulthood. Furthermore, breastfeeding can have positive effects on the mother, including promoting uterine involution and reducing the risk of cancer, osteoporosis, diabetes, and so on [2]. The importance of breastfeeding, especially providing newborns with colostrum, has been emphasized by the World Health Organization (WHO) [3]. However, there are challenges for women who have undergone cesarean delivery, such as postoperative pain, feeding techniques, and exposure to medications, which may affect their willingness to initiate breastfeeding immediately after surgery. According to the Chinese Guidelines for the Clinical Application of Antimicrobial Drugs (2015 edition), clinicians chose to use second-generation cephalosporins, either in combination with or without ornidazole, as the standard dosing regimen to prevent infection in mothers who have undergone a caesarean section or experienced premature rupture of the amniotic membrane. The issue of newborns being exposed to common perioperative drugs (e.g., nitroimidazoles) through breast milk and the potential for negative effects has been a significant concern for both clinicians and breastfeeding mothers.

Ornidazole is a highly effective agent that belongs to the third generation of nitroimidazole drugs and shares a similar molecular structure with metronidazole and tinidazole. Ornidazole has demonstrated excellent capabilities in combating protozoal infections, as well as anaerobic bacterial infections [4]. It is widely used in clinical settings for treating conditions such as amebiasis, giardia, trichomoniasis, and Crohn’s disease, as well as the prevention and treatment of various anaerobic bacterial infections. Adverse effects associated with ornidazole include digestive reactions (nausea, vomiting, hepatitis), neurological reactions (dizziness, drowsiness, peripheral nerve abnormalities), allergic reactions (rash, erythema), disulfiram-like reaction, and phlebitis [5]. Previous pharmacokinetic studies have shown that ornidazole is highly tissue permeable and broadly distributed throughout the body with a plasma protein binding rate of 11–13%. It has an elimination half-life of 11–14 h, which is approximately 1.7 times that of metronidazole [6,7,8]. Ornidazole is mainly metabolized in the liver through glucuronidation, with other metabolic pathways including oxidation and hydrolysis [9]. The majority of the prodrug and its metabolites are eliminated by the kidneys and excreted in the urine (about 70%), and only a small fraction is excreted in the bile and feces (about 20%) [10].

Given the significant secretion of nitroimidazoles through breast milk and the considerable variation in milk concentrations [11], it is essential to investigate the ornidazole exposure through breast milk to ensure the safety of lactation during or after drug administration in breastfeeding women. The milk concentration and the area under the concentration–time curve are commonly used indicators to assess drug exposure in breast milk. To obtain a more comprehensive understanding of potential risks to breastfed infants, we need to introduce several additional parameters associated with drug transfer into breast milk: the milk-to-plasma ratio, the absolute infant dose (AID), and the relative infant dose (RID) [12]. The AID represents the total amount of drug consumed by an infant through breast milk in a day, while the RID is a percentage derived by comparing the theoretical infant dose with the weight-adjusted maternal dose [13]. According to the WHO, medications that have an RID of less than 10% are deemed to be relatively safe for infants. On the other hand, breastfeeding women are advised to steer clear of medications that have an RID of greater than 25% [14,15]. A systematic study found the highest infant risk for metronidazole (11%) through the calculation and comparison of the percentages of AID to infant therapeutic doses for 20 antimicrobial drugs [16]. Although the effects of ornidazole on infants during breastfeeding are not well understood, it could be assumed that there may be a relatively high risk of exposure to infants due to its similar chemical structure and properties to metronidazole. Therefore, it is crucial to predict the amount of ornidazole that may be present in breast milk in order to minimize any potential effects on the newborn.

Preliminary studies have already been conducted to analyze the exposure and pharmacokinetic properties of metronidazole and tinidazole in the breast milk of lactating women [17,18,19]. However, there is a lack of information regarding the population pharmacokinetics (popPK) characteristics and excretion kinetics of ornidazole in this population. The main question to be addressed in this paper is when to begin breastfeeding after perioperative use of ornidazole in women undergoing caesarean section in order to ensure prolonged breastfeeding while minimizing the risk of drug excretion into the colostrum. In summary, we propose to use modeling and simulating approaches to analyze the pharmacokinetic behavior of ornidazole in the plasma and breast milk of lactating women, quantify the effects of various physiological factors on pharmacokinetic parameters, and assess the safety of breastfeeding after drug administration. This study will provide valuable insights for clinical guidance on lactation in perioperative women.

## 2. Materials and Methods

### 2.1. Patients and Study Design

This prospective trial enrolled pregnant women aged ≥ 18 years who were prescribed intravenous ornidazole for prevention or treatment of infections after cesarean delivery. According to the instructions on the label, the patients received 1 g of ornidazole intravenously 1 to 2 h before the procedure, followed by 0.5 g each at 12 and 24 h after the caesarean section. Mothers who refused to breastfeed or had severe hepatic or renal impairment were excluded from the current study. Approval was granted by the Ethics Committee of Wuhan Children’s Hospital (approval number: 2021R141-E01), and our study was conducted in compliance with the Declaration of Helsinki.

Usually, colostrum is produced in the first two to four days postpartum, and during this period, both plasma and breast milk specimens were procured for analysis [20]. Plasma samples (one sample per patient) were drawn at the time of routine biochemical tests at approximately 7 a.m., and breast milk samples (two samples per patient) were collected from both breasts via manual expression or an electric pump at around 9 a.m. or 3 p.m. All vital data, including precise dosing and sampling times, as well as the time of delivery, were documented. The sampling time after the last dose (TAD) and the postpartum sampling time (PST) were meticulously calculated. Other information including maternal age, body weight (WT), blood urea nitrogen (BUN), serum creatinine concentration (SCR), uric acid (UA), serum cystatin C (Cys-C), total bilirubin concentration (TBIL), direct bilirubin (DBIL), alanine aminotransferase (ALT), aspartate aminotransferase (AST), and gamma-glutamyl transferase (GGT) were recorded on the study day. Additionally, the maternal creatinine clearance (CrCL) was calculated using the Cockroft–Gault formula [21,22].

### 2.2. Analytical Methods

All samples were immediately transferred to the clinical pharmacology laboratory and analyzed on the same day. The plasma and milk samples were pretreated using solid-phase extraction and protein precipitation methods, respectively. Specific procedures were followed for each sample type. In the case of plasma samples, 0.5 mL was added to an activated C18 extraction column (China, Tianjin, Agela Technologies, Cleanert ODS C18, 200 mg/3 mL), then treated with 1 mL of 0.9% saline to wash out the protein. Subsequently, 1 mL of 40% acetonitrile solution was utilized to elate the targets. For the milk samples, 0.5 mL was mixed with an equal volume of acetonitrile, vortexed thoroughly and left to freeze at −20 °C for 10 min and then centrifugated at 10,000× *g* 4 °C for another 10 min to obtain the supernatant. Then, the eluate or supernatant was collected and 20 μL aliquots were injected into the chromatographic system for analysis.

Ornidazole concentrations were determined using a simple and rapid high-performance liquid chromatography (HPLC) method with ultraviolet detection (318 nm). Separations were performed on an Agela Innoval C18 column (250 mm × 4.6 mm, 5 μm) at 30 °C with a mobile phase of 25% *v*/*v* acetonitrile that was pumped at 0.8 mL/min. This method was proved to be sensitive and selective for the quantification of ornidazole in plasma and milk (Appendix A). The calibration curves were linear over ranges of 0.1–100 mg/L in plasma and 0.05–20 mg/L in milk, respectively. The lower limit of quantification was 0.1 mg/L in plasma and 0.05 mg/L in milk. The precision, accuracy, and recovery for ornidazole assay in both plasma and breast milk were all within acceptable ranges. For the assay in plasma, the intra- and interday relative standard deviations (RSDs) were 2.3–4.6% and 4.3–6.2% at concentrations between 0.3 and 75 mg/L, respectively. The mean accuracy and recovery were 92.8–96.8% and 88.0–95.7% at the same concentration levels, respectively. For the assay in plasma, the intra- and interday RSDs were 2.9–3.2% and 3.7–6.5% at concentrations between 0.15–15 mg/L, respectively. The mean accuracy and recovery were 94.4–97.5% and 87.2–92.5% at the same concentration levels, respectively.

### 2.3. Pharmacokinetic Modeling

The nonlinear mixed-effects modeling of the ornidazole concentration–time dataset in plasma and breast milk was conducted using the Phoenix NLME software (Version 8.3.4, Pharsight Corporation, Mountain View, CA, USA). The modeling was executed via the first-order conditional estimation–extended least-squares (FOCE-ELS) method. Selection of the model primarily relied on the likelihood ratio test of minimum objective function value (OFV), diagnosis through goodness-of-fit plots, and estimation of pharmacokinetic parameters. The R program (version 4.3.0, http://www.r-project.org/, accessed on 4 June 2023) was used for graphical analysis during the modeling process.

Given the sparse data on ornidazole, a one- or two-compartment model was first utilized to characterize the plasma concentration data, and then an additional parameter MPRcon (the milk-to-plasma concentration ratio) was employed to describe the correlation between ornidazole concentrations in milk and plasma. The interindividual variabilities (IIV) of each pharmacokinetic parameter were estimated by exponential error models, and the residual unexplained variabilities (RUVs) were assessed using additive, proportional, or combined error models. 

After the development of the base model, we investigated demographic data related to maternal age and WT, renal function indicators including BUN, SCR, UA, Cys-C, and CrCL, as well as hepatic function markers such as TBIL, DBIL, ALT, AST, and GGT. The timing of sampling during the study (TAD or PST) was also explored as a potential covariate. To avoid colinearity among covariates, correlation analysis was conducted. A stepwise approach was then utilized to evaluate the impacts of these covariates on the base model. The significance levels of *p* < 0.05 and *p* < 0.01 were set in the forward inclusion step and backward elimination step, respectively. Power functions were employed to test continuous variables centered on the median value, while categorical variables were assigned values of 0 or 1 and tested using exponential functions.

### 2.4. Model Evaluation

The final model was both graphically and statistically validated using goodness-of-fit plots, bootstrap analysis, and the visual predictive check (VPC) method. The bootstrap procedure was implemented using 1000 resampled datasets, and the resulting parameters were summarized as medians with 95% confidence intervals (CI) to assess the final estimates. The VPC was executed by simulating 1000 subjects based on the final model, and the model performance was evaluated through a visual comparison of the distribution characteristics for observations and simulations.

### 2.5. Simulation and Calculation

To provide the ornidazole concentration–time profiles in both plasma and milk, the Monte Carlo simulation with 1000 virtual subjects was conducted for the current dosing regimen (one preoperative dose of 1000 mg and two postoperative doses of 500 mg each within 24 h). The area under the concentration–time curve over 24 h (AUC_24_) was calculated through the trapezoidal rule, and the milk-to-plasma ratio was estimated using the simulated concentrations (MPRcon) or AUC_24_ of ornidazole (MPRauc). The influences of significant covariates on ornidazole exposure or penetration into breast milk were explored by means of graphical representation. 

Then, AID (mg/kg/day) was calculated as the product of the average ornidazole concentration in breast milk and the daily volume of milk ingested by infants [23]. RID was expressed as the ratio between the absolute infant dose and weight-adjusted maternal dose (a total dose of 2000 mg for a 70 kg women). An RID criterion of less than 10% was used to estimate the safety threshold for ornidazole concentration in breast milk, which was subsequently used to assess the potential risk of drug exposure to breastfed newborns. Of note, the volume of colostrum was obtained using the following equation describing the weight-normalized human milk intake (WHMI) [24]:(1)$$\mathrm{WHMI}=160.39\times \frac{0.232}{0.232-0.00252}\times \left(e^{-0.00252t}-e^{-0.232t}\right)$$
where t represents the infant age in days, and WHMI is expressed in mL/kg/day.

## 3. Results

### 3.1. Patients’ Characteristics

The 77 recruited mothers had a median age of 30.0 (18.0–41.0) years and a median body weight of 64.0 (43.7–90.0) kg. All of these patients met the criteria for inclusion, and none were excluded. In total, the participants contributed 87 plasma samples and 123 milk samples for analysis. The median values of plasma and milk concentration were 4.97 (range 0.33–20.00) mg/L and 1.43 (0.13–14.54) mg/L, respectively. The observations in plasma and breast milk are shown in Figure 1. The demographic and clinical characteristics of the mothers who participated in the study are summarized in Table 1.

### 3.2. Population Pharmacokinetic Analysis

In the current study, we determined that the one-compartment model was suitable for describing the plasma concentration profile of ornidazole. Then, we incorporated the parameter MPRcon into this model to link the concentrations in plasma and breast milk. The structural model was parameterized in terms of the milk-to-plasma concentration ratio (MPRcon), the apparent clearance (CL), and the volume of distribution (V) in the plasma compartment. The equations used to represent the structural model are as follows:(2)$$\mathrm{dA}/\mathrm{dt}=-\mathrm{CL}\;\times {\;C}_{1}$$
(3)$$C_{2}=\;\mathrm{MPRcon}\;\times {\;C}_{1}$$
(4)$$C_{1}=\frac{A}{V}$$

In the above model, A is the amount of ornidazole administered, and C_1_ and C_2_ denote the ornidazole concentration in plasma and breast milk, respectively. In addition, the IIV on CL and MPRcon were estimated using exponential models, and the RUV for both plasma and milk concentration data were well modeled using proportional error models.

The covariate screening procedure is detailed in Table 2. After the forward inclusion and backward elimination steps, the influences of PST on MPRcon and TBIL on CL were retained in the final model, resulting in a marked decrease in OFV of 38.12 and 10.55, respectively. To further explore the temporal variation in MPRcon, alternative time-dependent models were tested in addition to the power model (Appendix A). In this case, a simple power model with an estimated exponent (ModelI) could represent these temporal changes in MPRcon appropriately. In addition, the limited data did not support the estimation of additional model parameters. The final model for ornidazole was expressed using the following equations, and the parameter estimates are shown in Table 3. The typical values for MPRcon and CL were 0.58 and 1.89 L/h, respectively.
(5)$$\mathrm{MPRcon}=\mathrm{TVMPRcon}\;\times {\left(\frac{\mathrm{PST}}{{\mathrm{Median}}_{\mathrm{PST}}}\right)}^{1.37}$$
(6)$$\mathrm{CL}=\mathrm{TVCL}\;\times {\left(\frac{\mathrm{TBIL}}{{\mathrm{Median}}_{\mathrm{TBIL}}}\right)}^{-0.17}$$
where TVMPRcon and TVCL are the population typical values of MPRcon and CL, respectively. Median_PST_ and Median_TBIL_ are the median values of PST and TBIL in all patients, respectively.

### 3.3. Model Evaluation

As shown in Figure 2, the GOF plots indicate that the final model was able to fit the ornidazole data in plasma and milk well. Table 2 displays the results of the bootstrap analysis. The median bootstrap estimates are in line with the final model estimates with a small bias of less than ±5%, revealing satisfactory robustness for the final model. Furthermore, the VPC plots in Figure 3 demonstrate that the model had reasonable prediction performance.

### 3.4. Simulation and Calculation

Two sets of simulations were carried out to investigate the effect of key variables on breast milk penetration and exposure to ornidazole. In the first set of simulations, the TBIL was fixed at a reference value of 17 μmol/L, and the milk-to-plasma ratios predicted from either the simulated concentration or AUC_24_ were plotted against PST (Figure 4). In addition to the simulated MPRcon, the Bayesian-corrected MPRcon and the locally weighted scatterplot smoothing (LOWESS) curve generated by fitting the data are also depicted in Figure 4A. The Bayesian-corrected value of MPRcon was calculated as the ratio of the measured milk concentration to the plasma concentration adjusted by the Bayesian method to the milk sampling time. It is evident that the MPRcon exhibits a nonlinear increasing trend with PST. Moreover, the LOWESS curve is in close agreement with the median line of the simulated data, and the 95% confidence interval of the simulations could largely cover the Bayesian-corrected observations. This suggested that the validated final model could be employed to predict trends in MPRcon variation with PST that may be observed in clinical settings. As shown in Figure 4B and Appendix A, the median value of MPRauc increased from 0.186 on day one to 1.31 on day four. Another set of simulations were stratified according to TBIL levels: normal group (TBIL < 17 μmol/L) and abnormal group (17 μmol/L < TBIL < 34 μmol/L). The simulated concentration profiles of ornidazole in plasma and breast milk for virtual patients are shown in Figure 5. For patients in the normal group (Figure 5A), the median time to reach peak concentration in plasma and milk was 25 h and 27 h postpartum, respectively. The median peak concentrations (2.5th–97.5th percentile) in plasma and milk were 27.53 (23.12–31.94) mg/L and 5.41 (1.82–18.19) mg/L, respectively. For patients in the abnormal group (Figure 5B), the median time to reach peak concentration in plasma and milk was 25 h and 33 h postpartum, respectively. The median peak concentrations (2.5th–97.5th percentile) in plasma and milk were 30.22 (25.67–34.56) mg/L and 6.21 (2.01–20.34) mg/L, respectively. The minimum distance between the safety threshold and the 97.5% percentile of milk concentration in the normal and abnormal groups was 24.90 mg/L and 21.36 mg/L, respectively.

## 4. Discussion

The presence of tissue trauma and blood seepage in the uterine cavity after caesarean section predisposes to the spread of bacteria to the uterine cavity. The surgical procedure could also disrupt the normal bacteria present in the reproductive tract, making it more susceptible to opportunistic infections. While there is no worldwide consensus on the use of antimicrobial drugs to prevent infection in patients undergoing caesarean sections, many experts believe that prophylactic use of these drugs can reduce the incidence of postoperative infections and the length of hospital stay for both elective and emergency caesarean deliveries. Due to concerns about the potential adverse effects of preoperative medication on the fetus, most hospitals here prefer to use postoperative administration of antimicrobial drugs to prevent incisional infections in caesarean section patients. It is worth noting that many antibiotics can be excreted into breast milk, and nitroimidazoles have a higher likelihood of entering breast milk due to their small molecular weight and high fat solubility [25]. Several pharmacokinetic studies have shown that peak concentrations of metronidazole or tinidazole in breast milk could reach levels as high as approximately 15 mg/L within 2 h after administration to a breastfeeding woman [19,26]. However, there is a lack of research on the way ornidazole behaves in the body and its milk concentration in breastfeeding mothers. Furthermore, there is ongoing debate regarding the effects of these drugs on infants. Some reports suggest that the use of metronidazole by breastfeeding mothers may cause allergic reactions or alter the normal flora in the neonatal gut. Therefore, it is important to approach the use of medications by breastfeeding mothers with caution and consider the appropriate timing for breastfeeding based on available criteria during the postpartum period.

In the special group of patients undergoing cesarean sections, it is crucial to take measures to prevent infection and minimize any negative impacts on the newborns due to the use of antimicrobial drugs. The initial step in assessing the risk of neonatal adverse effects is to estimate the amount of drug ingested by the infant. Studies with small sample sizes make it challenging to predict changes in drug exposure for breastfeeding women and infants, as well as the risk of infant drug exposure through breast milk. Although pharmacokinetic studies that utilize intensive sampling designs could provide more detailed information, conducting such studies in this specific population is difficult. In this circumstance, a popPK study offers a valuable strategy to address this issue: using sparse data to establish a model and then analyzing the exposure to breastfed infants through a simulation approach. Our primary aim in performing the modeling analysis was to predict the concentration–time profiles of ornidazole in breast milk from maternal drug concentration levels in the plasma. Hence, physiological or mechanistic models that involve multiple compartments may not be suitable for this situation. In the present study, a one-compartment model fit the maternal plasma data well. Considering the paucity of data on milk concentrations, we were unable to use a separate compartment to describe the distribution of ornidazole from blood to milk, and instead estimated the MPRcon as a model parameter to determine the relationship between plasma and milk concentrations. Although the final model may not accurately reflect the physiological conditions, it demonstrated robust predictive capabilities for pharmacokinetic parameters.

The amount of drug that enters the infant’s body via breast milk mainly depends on the maternal plasma concentration, the ability of the drug to be transported from breast milk, and the amount of milk consumed by the infant [27]. The results of the covariate analysis reveal that maternal ornidazole clearance and exposure were significantly influenced by TBIL, while the penetration of ornidazole from breast milk was associated with PST. Ornidazole primarily undergoes metabolism in the liver, so the influence of liver function on its clearance is apparent [28]. In fact, TBIL plays an important role in the Child–Pugh classification and is commonly recognized as an essential covariate in certain popPK analyses, since it could indicate alterations in liver function to some extent [29,30]. Moreover, as illustrated in Figure 4, the distribution trends of both the simulated and Bayesian-corrected values of MPRcon show a tendency to increase with PST in a nonlinear fashion. From a physiological point of view, due to the limited excretion of colostrum by mothers after delivery, there may be an accumulation effect when ornidazole is transferred from blood to milk, resulting in different clearances between the blood and milk compartments, and thus the decreasing trend of ornidazole concentration in milk would fall behind that in blood. This accumulation phenomenon may be one of the reasons why the milk-to-plasma ratio increases with time. We speculate that the changes in the milk-to-plasma ratio within four days postpartum may be related to the increased permeability of the mammary gland to small-molecule drugs due to maternal hormonal shifts in the postpartum phase. Alternatively, it could be attributed to variations in the composition of breast milk constituents (such as lipids and proteins) under different circumstances and over the course of time [31,32]. According to the findings from the covariate screening, we further examined the impact of maternal liver function on infant risk. The simulation results show that women with abnormal liver function (TBIL > 17 μmol/L) had higher ornidazole levels in plasma and milk than those with normal liver function (TBIL < 17 μmol/L), but the exposures through colostrum of lactating women from both groups were below the safety threshold. Therefore, it can be tentatively concluded that it is relatively safe for mothers receiving ornidazole for perioperative infection prevention to breastfeed their newborns. Of course, reconsideration may be necessary in patients with more severe hepatic impairment.

Conducting clinical trials in vulnerable populations, like lactating women, could be particularly challenging. This study had some limitations, such as a small sample size, which prevented a more complex description of the breast milk excretion kinetics of ornidazole. Additionally, due to insufficient sample volume, we were unable to determine the content of various components in the breast milk, including fat and protein. However, this study adds to our understanding of the variability in ornidazole excretion into colostrum and assesses the safety of breastfeeding after mothers have been prescribed anti-infective drugs through modeling and simulating techniques.

## 5. Conclusions

This study represents an assessment of colostral transfer and exposure to ornidazole using a modeling strategy. By simulating drug concentration profiles in maternal plasma and breast milk, we found that prophylactic use of ornidazole had little impact on the safety of breastfeeding in postpartum women. While these findings may contribute to postnatal breastfeeding promotion, further confirmation is required through pediatric data.

## Figures and Tables

**Figure 1 pharmaceutics-15-02524-f001:**
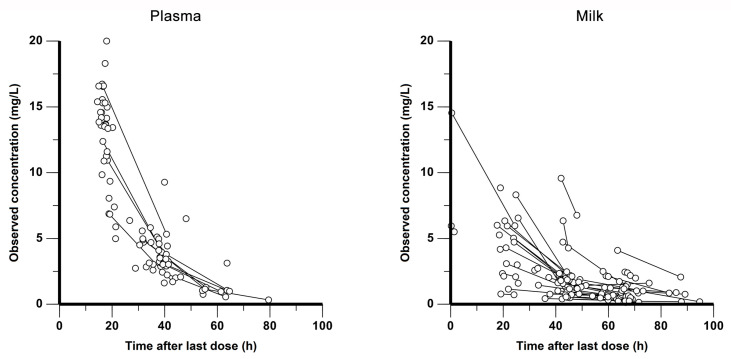
Concentration–time profiles of ornidazole in plasma (**left panel**) and breast milk (**right panel**). Ornidazole concentrations are shown as circles, and observations from the same subject are connected by short lines.

**Figure 2 pharmaceutics-15-02524-f002:**
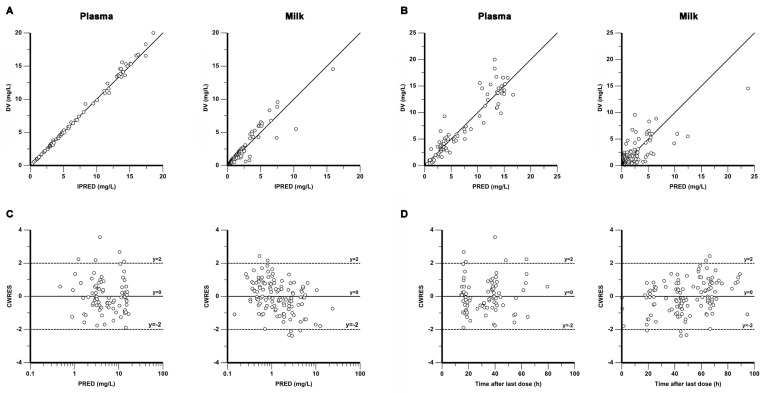
Goodness-of-fit plots of the final model for ornidazole data in plasma (left) and milk (right): (**A**) observations (DV) versus individual population predictions (IPRED), (**B**) DV versus population predictions (PRED), (**C**) conditional weighted residuals (CWRES) versus. PRED, (**D**) CWRES vs. time after dose (TAD).

**Figure 3 pharmaceutics-15-02524-f003:**
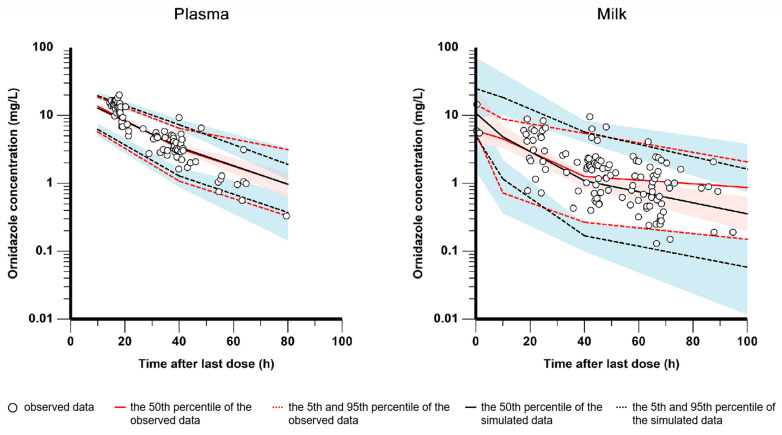
Visual predictive check of the final model for ornidazole concentrations in plasma (**left**) and milk (**right**): The black circles represent the observations. The dashed and solid red lines represent the 5th, 50th, and 95th percentile of the observed data, respectively. The dashed and solid black lines represent the 5th, 50th, and 95th percentile of the simulated data, respectively. The red and blue shaded areas show the 95% predicted intervals of the 5th–95th percentiles of the simulated data, respectively.

**Figure 4 pharmaceutics-15-02524-f004:**
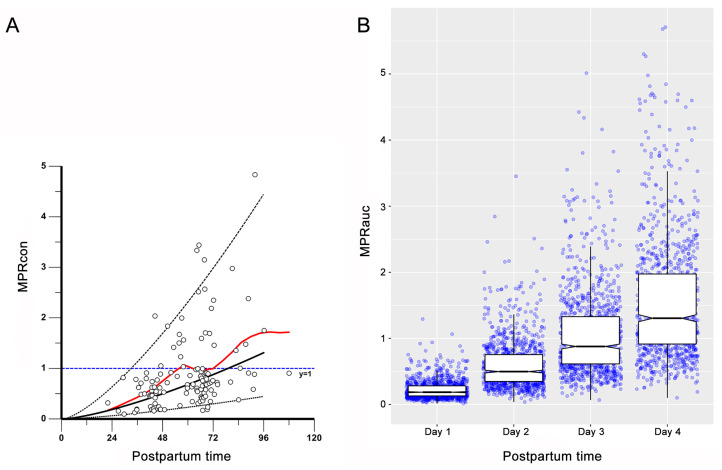
The simulated milk-to-plasma ratio determined through concentrations (**A**) or exposure (**B**) of ornidazole: In panel (**A**), the dashed and black solid lines represent the 2.5th, 50th, and 97.5th percentile of the predicted MPRcon, the blue dashed line represents the horizontal reference line (y = 1), the circles show the Bayesian-corrected measurements of MPRcon, and the red line represents the locally weighted scatterplot smoothing curve. In panel (**B**), the blue dots show the individual estimates of MPRauc, and the box shows the median, lower, and upper quartiles of the estimates.

**Figure 5 pharmaceutics-15-02524-f005:**
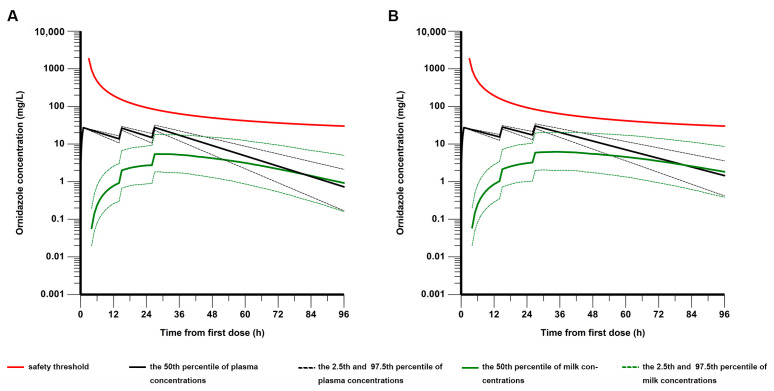
The simulated concentration profiles of ornidazole in plasma and breast milk for virtual patients classified by TBIL levels: (**A**) TBIL < 17 μmol/L and (**B**) 17 μmol/L < TBIL < 34 μmol/L. The black and green lines represent the 2.5th–97.5th percentile of plasma concentrations and milk concentrations, respectively. The red line indicates the safety threshold of milk concentration.

**Table 1 pharmaceutics-15-02524-t001:** Demographic and clinical characteristics of patients in this study (n = 77).

Characteristic	Number	Mean ± SD	Median (Range)
Patients	77		
Maternal age (years)		30.6 ± 4.3	30.0 (18.0–41.0)
WT (kg)		63.6 ± 9.9	64.0 (43.7–90.0)
Laboratory parameter			
BUN (mmol/L)		3.09 ± 0.73	2.90 (1.60–5.00)
SCR (μmol/L)		44.6 ± 7.9	45.0 (31.2–74.0)
UA (umol/L)		321.8 ± 78.9	308.0 (147.0–538.9)
CysC (mg/L)		1.25 ± 0.26	1.25 (0.70–2.60)
CrCL (mL/min)		185.51 ± 38.74	185.51 (108.52–307.12)
TBIL (μmol/L)		9.3 ± 3.5	9.1 (3.4–20.9)
DBIL (μmol/L)		2.7 ± 0.9	2.6 (0.9–6.2)
ALT (IU/L)		9.1 ± 5.1	8.0 (1.0–28.0)
AST (IU/L)		14.3 ± 4.9	14.0 (6.0–36.0)
GGT (IU/L)		11.4 ± 7.7	10.0 (3.0–46.0)

SD, standard deviation; WT, body weight; BUN, blood urea nitrogen; SCR, serum creatinine concentration; UA, uric acid; CysC, serum cystatin C; CrCL, maternal creatinine clearance; TBIL, total bilirubin concentration; DBIL, direct bilirubin; ALT, alanine aminotransferase; AST, aspartate aminotransferase; GGT, gamma-glutamyl transferase.

**Table 2 pharmaceutics-15-02524-t002:** Covariate screening and final model development process.

Step	Covariates Screening	OFV	ΔOFV	*p* Value	Comments
1	None	603.18	-	-	Base model
	Forward inclusion				
2	MPRcon-PST	565.06	−38.12	<0.05	
3	CL-WT	598.23	−4.95	<0.05	
4	CL-Age	602.93	−0.25	>0.05	
5	CL-CrCL	597.52	−5.66	<0.05	
6	CL-SCR	601.79	−1.39	>0.05	
7	CL-TBIL	592.63	−10.55	<0.05	
8	CL-ALT	603.14	−0.04	>0.05	
9	CL-AST	603.11	−0.07	>0.05	
10	CL-GGT	601.14	−2.04	>0.05	
11	MPRcon-PST, CL-TBIL	554.58	−10.48	<0.05	
12	MPRcon-PST, CL-WT	559.22	−5.84	<0.05	
13	MPRcon-PST, CL-CrCL	559.32	−5.74	<0.05	
14	MPRcon-PST, CL-TBIL-WT	548.68	−5.90	<0.05	Full model
	Backward elimination				
15	MPRcon-PST, CL-TBIL	554.58	5.90	>0.01	Final model

OFV, objective function value; ΔOFV, the change in the OFV; MPRcon, milk-to-plasma concentration ratio; PST, postpartum sampling time; CL, apparent clearance; WT, body weight; CrCL, maternal creatinine clearance; SCR, serum creatinine concentration; TBIL, total bilirubin concentration; ALT, alanine aminotransferase; AST, aspartate aminotransferase; GGT, gamma-glutamyl transferase.

**Table 3 pharmaceutics-15-02524-t003:** Parameter estimates and bootstrap results of the final model.

Parameters	Final Model		Bootstrap			Bias ^a^ (%)
	Estimate	RSE (%)	Median Estimate	2.5th Percentile	97.5th Percentile	
V (L)	35.75	4.89	35.77	31.99	39.81	0.06
CL (L/h)	1.89	2.78	1.89	1.78	2.00	0.11
MPRcon	0.58	8.63	0.58	0.48	0.68	0.02
θ_TBIL_	−0.17	−31.99	−0.16	−0.27	−0.07	−2.09
θ_PST_	1.37	15.32	1.38	0.95	1.80	0.47
Interindividual variability						
ω_CL_^2^	0.024	0.25	0.023	0.011	0.035	−4.21
ω_MPRcon_^2^	0.327	0.20	0.319	0.189	0.449	−2.48
Residual error (proportional error, %)						
σ_1_ (%)	8.18	13.01	8.02	5.67	11.30	−1.86
σ_2_ (%)	31.75	11.99	31.30	24.10	38.50	−1.30

RSE, relative standard error; V, volume of distribution; CL, apparent clearance; MPRcon, milk-to-plasma concentration ratio; θ_TBIL_, the effect of TBIL on CL; θ_PST_: the effect of PST on MPRcon; ω_CL_, square root of interindividual variance for CL; ω_MPRcon_, square root of interindividual variance for MPRcon; σ: residual variability of proportional error for plasma (σ_1_) and milk (σ_2_) concentration. Additionally, the IIV for V was omitted from the model due to a large amount of shrinkage. Note: ^a^ Bias = (median estimate from bootstrap analysis − estimate from the final model)/estimate from the final model.

## Data Availability

The data shown in this study are available on request from the corresponding author. The data are not publicly available due to ethical reasons as per local guidelines.

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
