# Peer review of "Ornidazole Transfer into Colostrum and Assessment of Exposure Risk for Breastfeeding Infant: A Population Pharmacokinetic Analysis"

_pharmaceutics, 2023, doi:10.3390/pharmaceutics15112524_

Round 1

Reviewer 1 Report

This will be a useful paper that provides valuable insight into a previously unstudied drug during breastfeeding. However, the study has one major flaw that needs to be corrected before publication. The authors used a static daily milk volume of 150 mL/kg as the infant intake. This number is appropriate after lactogenesis II, which occurs at about 7 days postpartum, but may not be fully reached for 2-3 weeks postpartum.[1-3] The authors allude to this on lines 357-358, but do not act on it appropriately. The authors are reporting on infants in the first 2-4 days postpartum when only colostrum is being produced. The volume of colostrum is markedly (by an order of magnitude) lower than 150 mL/kg/day. The calculation of infant exposure need to be scaled back based on milk volumes that are expected during this time. An example of how the increasing volume of colostrum can be incorporated into a model is the article by Yeung et al.[4] The good news is that if the appropriate volume is used to calculate the infant exposure of ornidazole in the first 4 days postpartum, drug exposure will be much less and provide more assurance of the safety of prophylactic ornidazole use.

References

1.           Kato I, Horike K, Kawada K, et al. The trajectory of expressed colostrum volume in the first 48 hours postpartum: An observational study. Breastfeed Med 2022;17:52-8.

2.           Yeung CHT, Fong S, Malik PRV, Edginton AN. Quantifying breast milk intake by term and preterm infants for input into paediatric physiologically based pharmacokinetic models. Matern Child Nutr 2020;16:e12938.

3.           Rios-Leyvraz M, Yao Q. The volume of breast milk intake in infants and young children: A systematic review and meta-analysis. Breastfeed Med 2023;18:187-97.

4.           Yeung CHT, Ito S, Autmizguine J, Edginton AN. Incorporating breastfeeding-related variability with physiologically based pharmacokinetic modeling to predict infant exposure to maternal medication through breast milk: A workflow applied to lamotrigine. AAPS J 2021;23:70.

Line 274 and Figure 4: Do the authors have any proposed explanation for an increasing MPR over the first 4 days postpartum? This is an unusual finding that is not addressed in the paper.

Rewording

The paper is generally well written, but a few word choices can be improved:

Line 42: change regeneration to “involution”

Line 99: change lactate to “begin breast feeding”

Line 109: change enrolls to “enrolled”

Line 117: change within to “in the first”

Line 123: change was to “were”

Line 196: this volume needs to be changed as indicated above

Line 197: delete “And” at the beginning of the sentence

Section 3.1 does not adequately describe several numbers

Line 204: is 30 years a mean or median age?

Line 205: is the maternal weight a mean or median?

Line 207: the values in parentheses should go in a new sentence, because this sentence just establishes what was donated and the values were not determined until after they were analyzed.

Line 221: delete “This” at the beginning of the sentence

Line 225: change devote to “denote”

Lines 226-227: “And k is the milk-to-plasma ratio calculated from concentration data.” This statement needs more explanation. The k values appear to be single time point values (rather than AUC values). If this is the case, please clearly state it.

Line 232: I would change remarkable to “marked”

Table 3 is confusing because the lines in the headings go across the entire table with no break. This makes it hard to know which values are part of the Final Model and which are part of the Bootstrap. There should be breaks in the bottom two lines where appropriate.

Line 308: Smaller begs the question, smaller than what? Maybe just “small” would be clearer.

Line 309: Greater poses the same issue. Is the lipid solubility of ornidazole “high”?

The paper is generally well written, but a few word choices can be improved. Suggestions are provided to authors.

Author Response

Comment 1: This will be a useful paper that provides valuable insight into a previously unstudied drug during breastfeeding. However, the study has one major flaw that needs to be corrected before publication. The authors used a static daily milk volume of 150 mL/kg as the infant intake. This number is appropriate after lactogenesis II, which occurs at about 7 days postpartum, but may not be fully reached for 2-3 weeks postpartum.[1-3] The authors allude to this on lines 357-358, but do not act on it appropriately. The authors are reporting on infants in the first 2-4 days postpartum when only colostrum is being produced. The volume of colostrum is markedly (by an order of magnitude) lower than 150 mL/kg/day. The calculation of infant exposure need to be scaled back based on milk volumes that are expected during this time. An example of how the increasing volume of colostrum can be incorporated into a model is the article by Yeung et al.[4] The good news is that if the appropriate volume is used to calculate the infant exposure of ornidazole in the first 4 days postpartum, drug exposure will be much less and provide more assurance of the safety of prophylactic ornidazole use.

References

  1. Kato I, Horike K, Kawada K, et al. The trajectory of expressed colostrum volume in the first 48 hours postpartum: An observational study. Breastfeed Med 2022;17:52-8.
  2. Yeung CHT, Fong S, Malik PRV, Edginton AN. Quantifying breast milk intake by term and preterm infants for input into paediatric physiologically based pharmacokinetic models. Matern Child Nutr 2020;16:e12938.
  3. Rios-Leyvraz M, Yao Q. The volume of breast milk intake in infants and young children: A systematic review and meta-analysis. Breastfeed Med 2023;18:187-97.
  4. Yeung CHT, Ito S, Autmizguine J, Edginton AN. Incorporating breastfeeding-related variability with physiologically based pharmacokinetic modeling to predict infant exposure to maternal medication through breast milk: A workflow applied to lamotrigine. AAPS J 2021;23:70.

Response: Thank you for your valuable comments. At your suggestion, we used the WHMI equation from the study by Yeung et al. to calculate colostrum intake during the first four days postpartum. A new safety threshold was then created and plotted as a curve over time in Figure 5. Since the volume of colostrum during the first four days postpartum is much less than the expected value of 150 ml/kg/day, the exposure to ornidazole in breast milk can be predicted to be below the safety threshold. Therefore, it can be tentatively concluded from the simulation results that breastfeeding after ornidazole prophylaxis is relatively safe in mother with normal or abnormal liver function. The relevant content in the Title, Abstract, Methods, Results, Discussion and Conclusions sections have been revised.

Comment 2: Line 274 and Figure 4: Do the authors have any proposed explanation for an increasing MPR over the first 4 days postpartum? This is an unusual finding that is not addressed in the paper.

Response: We found a significant effect of postpartum time on MPR during covariate screening analysis and therefore speculated that the changes in MPR within four days postpartum might be related to the increased permeability of the mammary gland to small molecule drugs due to maternal hormonal shifts in the postpartum phase. Alternatively, it could be attributed to variations in the composition of breast milk constituents (such as lipids and proteins) under different circumstances and over the course of time. These explanations could be found in the Discussion section. (Line 362-366)

Rewording

The paper is generally well written, but a few word choices can be improved:

Line 42: change regeneration to “involution”

Response: Thank you for your careful work, we followed this suggestion.

Line 99: change lactate to “begin breast feeding”

Response: We followed this suggestion.

Line 109: change enrolls to “enrolled”

Response: We are so sorry to make this mistake. We have corrected it in the new manuscript.

Line 117: change within to “in the first”

Response: We followed this suggestion.

Line 123: change was to “were”

Response: We followed this suggestion.

Line 196: this volume needs to be changed as indicated above

Response: We followed this suggestion. The volume of colostrum intake during the first four days postpartum was calculated using the WHMI equation from the study by Yeung et al. (Line 207-208)

Line 197: delete “And” at the beginning of the sentence

Response: We followed this suggestion.

Section 3.1 does not adequately describe several numbers

Line 204: is 30 years a mean or median age?

Response: This value is the median age, we have rephrased the sentence. (Line 213)

Line 205: is the maternal weight a mean or median?

Response: This value is the median weight, we have rephrased the sentence. (Line213)

Line 207: the values in parentheses should go in a new sentence, because this sentence just establishes what was donated and the values were not determined until after they were analyzed.

Response: We followed this suggestion.

Line 221: delete “This” at the beginning of the sentence

Response: Thank you very much, we have corrected this mistake.

Line 225: change devote to “denote”

Response: We have corrected this mistake.

Lines 226-227: “And k is the milk-to-plasma ratio calculated from concentration data.” This statement needs more explanation. The k values appear to be single time point values (rather than AUC values). If this is the case, please clearly state it.

Response: The parameter k represents the milk-to-plasma concentration ratio (Cmilk/Cplasma) and is used to establish the relationship between concentration data in plasma and milk. We clarified this in the new version of manuscript. (Line 235-236)

Line 232: I would change remarkable to “marked”

Response: We followed this suggestion.

Table 3 is confusing because the lines in the headings go across the entire table with no break. This makes it hard to know which values are part of the Final Model and which are part of the Bootstrap. There should be breaks in the bottom two lines where appropriate.

Response: Following your suggestion, we added spaces between the two columns in Table 3 to distinguish between final model estimates and bootstrap estimates.

Line 308: Smaller begs the question, smaller than what? Maybe just “small” would be clearer.

Response: We followed this suggestion.

Line 309: Greater poses the same issue. Is the lipid solubility of ornidazole “high”?

Response: We followed this suggestion.

Reviewer 2 Report

The study developed a model to assess the transfer of orinidazole into colostrum. Population pharmacokinetic analysis was conducted. The profile of ornidazole in the plasma and the breast milk was predicted using the model. The model also simulated orinidazole concentration in milk under the impact of abnormal liver function on the orinidazole concentration. The study showed some interesting results. The following concerns need to be addressed:

  1. Many labels, lines, and colors are used in Figure 3. Plot legends need to be added to indicate which are from observed data and which are from the simulated data.
  2. Please also add legends to Figure 5.
  3. Figure 4 labels (A) and (B) on the plot are missing.
  4. Please correct the subscript of "AUC24" in Line 192 on page 4.

Author Response

The study developed a model to assess the transfer of orinidazole into colostrum. Population pharmacokinetic analysis was conducted. The profile of ornidazole in the plasma and the breast milk was predicted using the model. The model also simulated orinidazole concentration in milk under the impact of abnormal liver function on the orinidazole concentration. The study showed some interesting results. The following concerns need to be addressed:

Comment 1: Many labels, lines, and colors are used in Figure 3. Plot legends need to be added to indicate which are from observed data and which are from the simulated data.

Response: Thanks for your suggestion, legends were plotted in the new Figure 3.

Comment 2: Please also add legends to Figure 5.

Response: Thanks for your suggestion, legends were plotted in the new Figure 5.

Comment 3: Figure 4 labels (A) and (B) on the plot are missing.

Response: Thank you very much, we have corrected this mistake in the new Figure 4.

Comment 4: Please correct the subscript of "AUC24" in Line 192 on page 4.

Response: Thank you very much, we have corrected this mistake.

Reviewer 3 Report

Pharmacokinetic characteristics of ornidazole in the plasma and breast milk of lactating women was evaluated using modeling and simulating approaches, and its related physiological factors were quantified. A population pharmacokinetic model was established for prediction of ornidazole exposure in milk and determination of a safe period for mothers to breastfeed after taking medications. The study is fully investigated, and the manuscript is well organized. Before its acceptance, some minor doubts could be well addressed, as follows,

1.     A number of plasma and bread milk samples were obtained from mothers after the administration of ornidazole, which was quantified by an HPLC method. LLOQ values of the HPLC method for ornidazole in plasma and bread milk were reported. However, either the HPLC method was not published previously or the validated results of the method were not shown in the manuscript. In addition, the specificity chromatogram is should be shown in case the HPLC is an in-house method.

2.     Correct some grammatical errors, like line 32, lines 49-50, line 67, line 110.

Correct some grammatical errors, like line 32, lines 49-50, line 67, line 110.

Author Response

Pharmacokinetic characteristics of ornidazole in the plasma and breast milk of lactating women was evaluated using modeling and simulating approaches, and its related physiological factors were quantified. A population pharmacokinetic model was established for prediction of ornidazole exposure in milk and determination of a safe period for mothers to breastfeed after taking medications. The study is fully investigated, and the manuscript is well organized. Before its acceptance, some minor doubts could be well addressed, as follows,

Comment 1: A number of plasma and bread milk samples were obtained from mothers after the administration of ornidazole, which was quantified by an HPLC method. LLOQ values of the HPLC method for ornidazole in plasma and bread milk were reported. However, either the HPLC method was not published previously or the validated results of the method were not shown in the manuscript. In addition, the specificity chromatogram is should be shown in case the HPLC is an in-house method.

Response: Thank you for your suggestion. The validated results of the method were described in detail in the relevant section (2.2. Analytical methods) and the specificity chromatograms were provided in the supplementary material (Figure S1). (Line 145-156)

Comment 2: Correct some grammatical errors, like line 32, lines 49-50, line 67, line 110.

Response: Thank you for your comments, we have corrected these errors in the new version of the manuscript.

Round 2

Reviewer 1 Report

The changes made to the manuscript are adequate. The new milk volume calculation considerably improves the paper 

Author Response

Thank you very much for your careful review and constructive suggestions with regard to our manuscript.